# A Chaotic Dynamics Framework Inspired by Dorsal Stream for Event Signal Processing

**Yu Chen** [1 2]  **Jing Lian** [3]  **Zhaofei Yu** [4]  **Jizhao Liu**[† 1]  **Jisheng Dang**[† 1]  **Gang Wang**[† 2]

## Abstract

Event cameras are bio-inspired vision sensors that encode visual information with high dynamic range, high temporal resolution, and low latency. Current state-of-the-art event stream processing methods rely on end-to-end deep learning techniques. However, these models are heavily dependent on data structures, limiting their stability and generalization capabilities across tasks, thereby hindering their deployment in real-world scenarios. To address this issue, we propose a chaotic dynamics event signal processing framework inspired by the dorsal visual pathway of the brain. Specifically, we utilize Continuous-coupled Neural Network (CCNN) to encode the event stream. CCNN encodes polarity-invariant event sequences as periodic signals and polarity-changing event sequences as chaotic signals. We then use continuous wavelet transforms to analyze the dynamical states of CCNN neurons and establish the high-order mappings of the event stream. The effectiveness of our method is validated through integration with conventional classification networks, achieving state-of-the-art classification accuracy on the N-Caltech101 and N-CARS datasets, with results of 84.3% and 99.9%, respectively. Our method improves the accuracy of event camera-based object classification while significantly enhancing the generalization and stability of event representation. Our code is available in https://github.com/chenyu0193/ACDF.

## 1. Introduction

Event cameras, inspired by the three-layer structure of the peripheral retina in primates, are neuromorphic sensors designed for silicon-based vision. Each pixel of the event sensor operates independently and continuously to detect changes in light intensity within a scene. When the change exceeds a preset threshold, an event signal is triggered. The event signal encodes spatiotemporal information, including a timestamp, spatial coordinates, and polarity. The unique sampling mechanism enables event stream data with the advantages of high temporal resolution, low redundancy, a wide dynamic range, and minimal latency. However, frame-based vision algorithms designed for image sequences are not directly applicable to event data (Gallego et al., 2020).

To address this challenge, event streams are compressed event into frames, producing 2D event frame images via frequency accumulation methods (Gallego et al., 2018; Stoffregen & Kleeman, 2019; Gallego et al., 2019; Almatrafi et al., 2020; Brebion et al., 2021; Hagenaars et al., 2021; Paredes-Vallés & De Croon, 2021; Shiba et al., 2022b). Similarly, a timestamp-based event representation method, known as event surfaces (Mueggler et al., 2017; Lagorce et al., 2017; Sironi et al., 2018), updates the latest timestamp information to capture motion changes in the spatial positions of objects. To more effectively leverage the spatiotemporal information within event streams, a spatiotemporal histogram-based representation was proposed, called voxel grids (Zhu et al., 2018; Deng et al., 2022; Xie et al., 2022; Baldwin et al., 2022). This method discretizes the time domain and employs linearly weighted accumulation to allocate events into corresponding voxels.

With advancements in deep learning and large-scale computation, data-driven, end-to-end neural networks (Gehrig et al., 2019; Sekikawa et al., 2019; Wang et al., 2019; Bi et al., 2019; Cannici et al., 2020; Yang et al., 2019; Bi et al., 2020; Deng et al., 2021; Schaefer et al., 2022; Sabater et al., 2022; Wang et al., 2022) have gained increasing popularity. These methods efficiently exploit the asynchronous spatiotemporal characteristics of event streams. Additionally, bio-inspired spiking neural networks (Fang et al., 2021; Li et al., 2022; Shen et al., 2023; Wang et al., 2024) simulate biological spike signal processing mechanisms, integrat-

---

[1]School of Information Science and Engineering, Lanzhou University, Lanzhou 730000, China [2]NAIVE Lab, Brain Research Center, Beijing Institute of Basic Medical Sciences, Beijing 100850, China [3]School of Electronics and Information Engineering, Lanzhou Jiaotong University, Lanzhou 730070, China [4]School of Institute for Artificial Intelligence, Peking University, Beijing 100850, China. Correspondence to: Jizhao Liu <liujz@lzu.edu.cn>, Jisheng Dang <dangjsh@mail2.sysu.edu.cn>, Gang Wang <g_wang@foxmail.com>.

*Proceedings of the 42nd International Conference on Machine Learning*, Vancouver, Canada. PMLR 267, 2025. Copyright 2025 by the author(s).

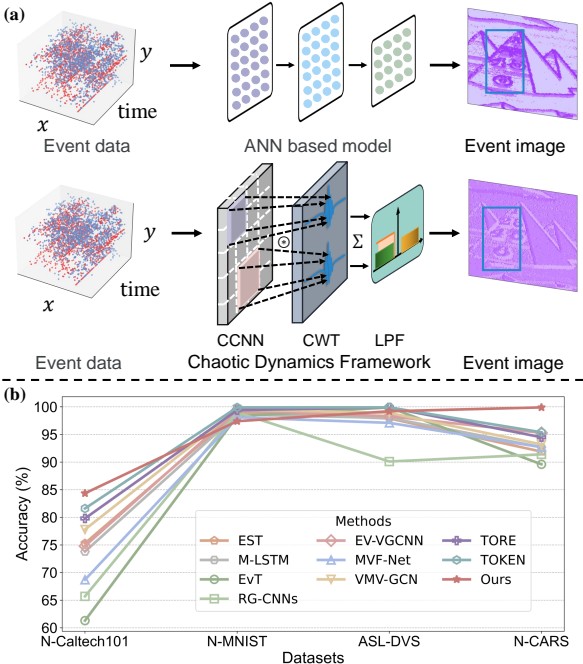

*Figure 1.* **The architecture and performance of the proposed framework.** (a) The chaotic dynamic framework, where event stream data are input into CCNN neurons in the form of coordinates. The dynamic characteristics of neurons are analyzed using CWT, and the event representation is subsequently obtained through LPF. (b) Comparative evaluation of the model's classification performance on multiple datasets. The proposed method achieves superior classification accuracy across diverse datasets, demonstrating its strong generalization capability and robust adaptability in processing event-based data.

ing pulse signals and sampling events based on their firing times once a predefined threshold is exceeded. These event representation methods are shown in Figure 2.

However, the above event representation methods exhibit significant limitations in terms of generalization and stability, particularly in their inconsistent performance across datasets, lack of robustness, and strong dependency on specific representational structures. Specifically, these methods often struggle with scenarios involving sparse data or high dynamic range, highlighting their lack of robustness and adaptability. Moreover, most event representation methods are designed to specific representational frameworks, such as spatiotemporal voxel grids or local features, making them less adaptable to diverse event data. These obstacles hinder the widespread applicability of existing methods in real-world scenarios.

Drawing inspiration from the mechanisms of the brain for processing visual information offers a promising approach to overcoming current obstacles. Recent experimental studies have demonstrated that neural responses in visual processing exhibit consistent patterns, even while adapt-

ing to diverse stimuli and experimental conditions (Groen et al., 2022; Gong et al., 2023). The experimental evidence strongly supports the brain's remarkable ability to maintain stable and generalizable visual processing across different datasets and conditions. By effectively harnessing the neural processing mechanisms of the visual cortex, we can advance the development of event representation methods with improved generalization and stability.

In this work, we introduce a continuous-coupled neuron network (CCNN) inspired by the primary visual cortex (Liu et al., 2022). The CCNN exhibits electrophysiological properties similar to those of mammalian neuron clusters, generating periodic sequence outputs in response to constant input signals and chaotic sequence outputs in response to varying signals. This input-output behavior enables the network to effectively distinguish between stable events, characterized by constant polarity patterns, and dynamic events, characterized by varying polarity patterns. Leveraging the unique characteristics of the CCNN, we separate constant-polarity events from varying-polarity events within the same sampling period. The separated event sequences are then processed using continuous wavelet transforms (CWT) to extract spatiotemporal information, establishing a high-order mapping from the event stream to event frames. The overall architecture and performance of the framework are shown in Figure 1. To further enhance the accuracy of motion extraction, the framework integrates a deep neural network to achieve precise localization and recognition of moving objects, as shown in Figure 3.

The main contributions of this work are summarized as follows:

- We propose an event stream processing framework inspired by the brain's dorsal visual pathway. We introduce the spatial-temporal information encoding mechanism of the brain's dorsal pathway, also known as the "where" pathway, into the event stream data processing framework, effectively establishing a high-order mapping from event streams to event frames.

- This framework utilizes CCNN to encode constant-polarity event sequences as periodic signals and varying-polarity event sequences as chaotic signals, effectively achieving robust event representation. When combined with traditional deep neural networks, the framework successfully performs object classification for event cameras.

- The proposed framework is evaluated on multiple datasets, achieving state-of-the-art accuracy on specific benchmarks. It also demonstrates competitive performance across a variety of datasets. The results demonstrate the framework's strong generalization across different data structures.

## 2. Related Work

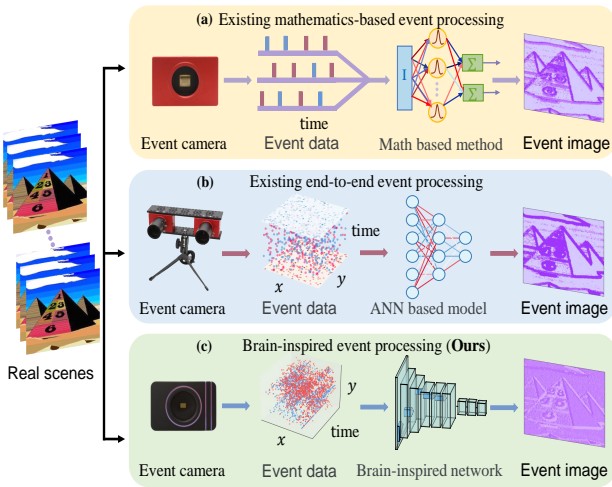

*Figure 2.* **The review of representation of asynchronous events.** Existing methods can be classified into three categories: mathematical-based methods, end-to-end processing models based on ANNs, and brain-inspired networks.

### 2.1. The Frequency Accumulation Methods

Prior event representations were primarily task-specific methods based on mathematical models. For instance, the surface of active events (SAE) encodes three-dimensional event streams into frame images based on timestamps to capture event motion trajectories, demonstrating promising results in corner detection (Mueggler et al., 2017). Similarly, the concept of the "distance surface" was introduced, where pixel intensities are derived as proxies based on the distances of event points to motion edges, and applied to optical flow estimation (Almatrafi et al., 2020). Building on the "distance surface," inverse exponential calculations were incorporated, resulting in a novel dense "inverse exponential distance surface" representation that addresses noise sensitivity and unbounded influence regions (Brebion et al., 2021). However, these event representations typically adopt a frequency accumulation approach, often resulting in blurred event edges. Therefore, event alignment is required during the process of converting event streams into event frames.

### 2.2. The Contrast Maximization Methods

Contrast maximization serves as an effective method to address image blurring. It maximizes an evaluation function to assess the alignment of event edges caused by object motion, resulting in clear event frame images through motion estimation. Based on this, a unifying contrast maximization framework is proposed for motion, depth, and optical flow estimation with event cameras. (Gallego et al., 2018). Subsequently, researchers classified and investigated reward

functions affecting event alignment, exploring the impact of different evaluation methods on recovering sharp event frames and their performance across various applications (Gallego et al., 2019) (Stoffregen & Kleeman, 2019). Beyond generating aligned event images, contrast maximization has also been employed in deep learning as a form of supervision. For instance, The contrast maximization-based self-supervised learning framework has achieved competitive results in optical flow estimation (Hagenaars et al., 2021; Paredes-Vallés & De Croon, 2021; Shiba et al., 2022b). However, the contrast objective (variance) may overfit the events, which can push the events to accumulate in too few pixels (event collapse (Shiba et al., 2022a)).

### 2.3. The Deep Learning-based Methods

With the increase in computility, deep learning has rapidly become the dominant approach in asynchronous event data representation. The Event Spike Tensor (EST) is the first data-driven, end-to-end event representation method. It utilizes a multilayer perceptron (MLP) to learn the optimal mapping function, thereby maximizing task performance (Gehrig et al., 2019). Matrix-LSTM replaces the MLP with an LSTM, leveraging temporally accumulated pixel information to construct a 2D event representation, thereby further optimizing the learning framework (Cannici et al., 2020). To fully leverage the sparsity and asynchronicity of event data, graph-based representation methods utilizing graph neural networks have been proposed (Xu et al., 2018; Bi et al., 2020; Schaefer et al., 2022; Deng et al., 2022; Wang et al., 2024). These methods process event data in the form of a temporally evolving graph, efficiently maintaining both sparsity and high temporal resolution. Dense event representations based on convolutional neural networks (CNNs) achieve superior task performance; however, they are computationally intensive, which limits their practical deployment. In contrast, sparse event representations leveraging graph neural networks (GNNs) enhance efficiency by exploiting the spatiotemporal characteristics of asynchronous events, although they typically exhibit lower accuracy and are constrained in terms of application scope.

## 3. Methods

### 3.1. Event Field

The event streams generated by event cameras can be considered as point sets in three-dimensional space, where each event point is represented as four-dimensional data. The event point consists of spatial coordinates $x$ and $y$, polarity, and timestamps. Inspired by (Gehrig et al., 2019), this point set is represented by the following equation:

$$E(x,y,p,t) = \sum_{e_n \in \varepsilon} \delta(x - x_n, y - y_n, p - p_n)\delta(t - t_n). \quad (1)$$

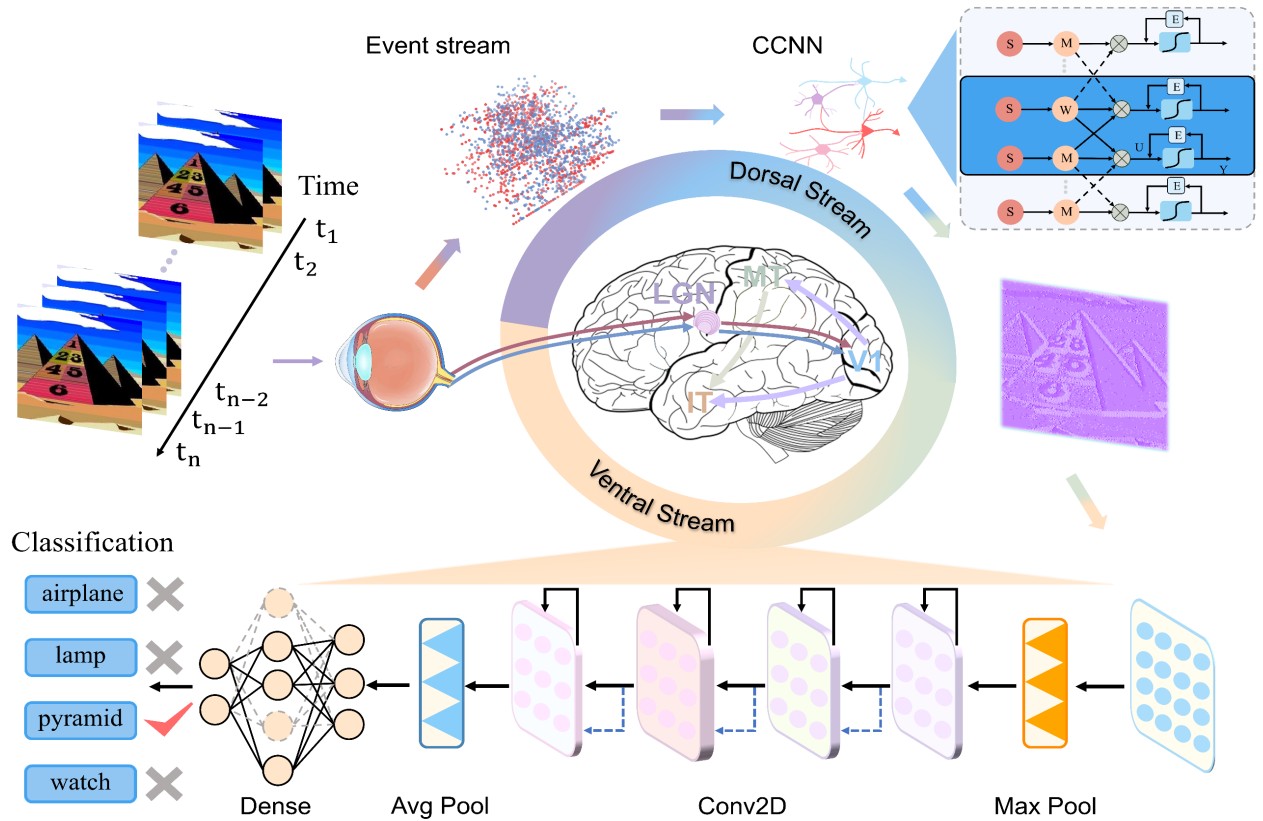

*Figure 3.* **The human brain's visual cortex recognizes moving objects through the dorsal and ventral pathways.** Event cameras mimic the three-layer structure of the peripheral retina in humans, encoding moving objects into event stream data. We utilize a chaotic dynamics framework, based on CCNN, to map the event stream data into event representations inspired by the dorsal stream. Subsequently, the event representations are sent from the MT area to the IT area, where recognition of moving objects is achieved through multiple layers of neural networks.

The event point set is continuously represented by the function $E(x, y, p, t)$. Each event point in the event stream is represented by $\delta(\cdot)$ to capture its spatiotemporal information and polarity. In three-dimensional space $\varepsilon$, an event point $e_n = (x_n, y_n, p_n, t_n)$ generates a Dirac impulse when its spatiotemporal information and polarity match. This indicates the occurrence of the event.

### 3.2. Dorsal Pathway-Inspired Event Representation

**Sampling.** When processing three-dimensional event stream data, sampling is the primary step and can generally be categorized into two categories: fixed sampling and adaptive sampling. The sampled event bins are expressed as follows:

$$E[x_i, y_j, p_k, t_l] = \sum_{e_k \in \varepsilon} \delta(x_i - x_n, y_j - y_n, p_k - p_n)$$
$$\delta(t_l - t_n), \qquad (2)$$

where $x_i \in \{0, 1, 2, \cdots X\}, y_j \in \{0, 1, 2, \cdots Y\}$ represent the resolution of the event frame, $p_k \in \{-1, 1\}$ denotes the polarity of the event. $t_k \in \{t_0 + N(\eta \cdot \Delta t)\}$, where $t_0$ is the starting time, $N$ is the number of event bins, $\Delta t$ is the time interval, and $\eta$ is the adjustment factor. For the fixed sampling method, $\eta$ remains constant, whereas for adaptive sampling, $\eta$ is adjusted dynamically based on specific requirements.

Each event point in the event stream data contains the position coordinates $(x, y)$ of the moving object, a timestamp $t$, and polarity $p$. In this work, the coordinate information is designated as the key, while the polarity sequence corresponding to the same coordinate serves as the value. After aligning all values, they are uniformly input into the CCNN. Different polarity variation sequences result in different types of output signals.

$$V(e_n) = F((x_i, y_j), \varepsilon)$$
$$= \{(p_k, t_l)|E[x_i, y_j, p_k, t_l] \in \varepsilon\}, \qquad (3)$$

where $F(\cdot)$ represents the mapping function, $(x_i, y_j)$ de-

notes the spatial coordinates used as keys, and $V(e_k)$ represents the polarity timestamp sequence corresponding to the coordinates, used as values.

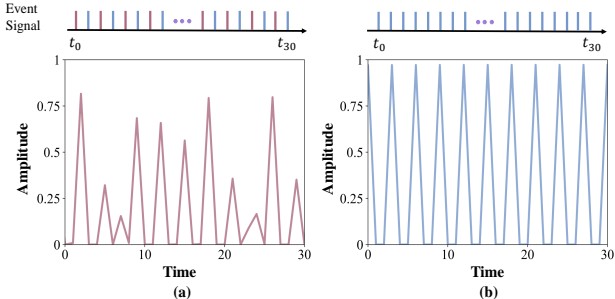

Figure 4. **The input-output characteristics of the CCNN.** (a) When stimulated by event signals with changing polarity, the CCNN generates chaotic sequences as output. (b) When stimulated by event signals with constant polarity, the CCNN produces periodic sequences as output.

**Continuous-coupled Neural Network.** When inputting all mapped polarity sequences within the same sampling period into the CCNN, a non-coupled CCNN is chosen for simplicity. Its mathematical model is described as shown in the following equation:

$$U(k) = e^{-\alpha_f} U(k-1) + V(e_k)$$
$$Y(k) = \frac{1}{1 + e^{-(U(k)-E(k))}} \qquad (4)$$
$$E(k) = e^{-\alpha_e} E(k-1) + V_E Y(k-1),$$

where $U$ is an independent variable influenced solely by the external input $V$, which in this work corresponds to the polarity sequence at a specific coordinate.

When the polarity varies uniformly, the general term equation of $U(k)$ is expressed as:

$$U(k) = V \cdot \frac{1 - e^{-k\alpha_f}}{1 - e^{-\alpha_f}}. \qquad (5)$$

Through derivation, the expression for period $k$ is obtained as:

$$k = 1 + \frac{1}{\alpha_f} ln \frac{V}{V - (1 - e^{-\alpha_f})(E(0) - \ln\left(\frac{V_E}{(1-e^{-\alpha_e})E(0)} - 1\right))}. \qquad (6)$$

Thus, CCNN neurons output a periodic sequence $Y(k)$ under constant stimulation, with the frequency of the period determined by the intensity of the input stimulus.

When the polarity varies periodically, $V(e_k) = \{0, 1, 0, 1, \cdots 0, 1\} = -\frac{i^k + (-i)^k}{2} = \sin\left(\frac{k \cdot \pi}{2}\right)$. The general term equation for $U(k)$ is given as:

$$U(k) = \frac{e^{-k\alpha_f} \sin\left(\frac{k \cdot \pi}{2}\right) - \alpha_f e^{-k\alpha_f} \cos\left(\frac{k \cdot \pi}{2}\right)}{1 + \alpha_f^2}. \qquad (7)$$

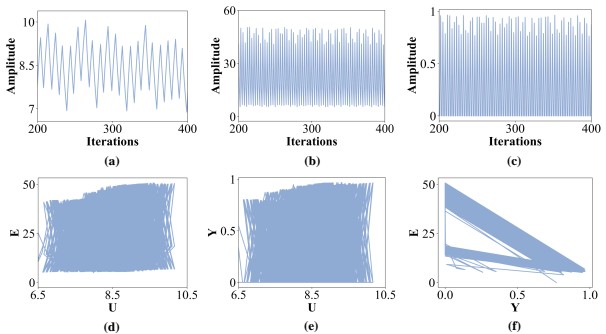

Figure 5. **Waveform and phase space plot of CCNN neuron.** (a) Waveform of U. (b) Waveform of E. (c) Waveform of Y. (d) Phase space plot of U-E plane. (e) Phase space plot of U-Y plane. (f) Phase space plot of E-Y plane.

According to equation (4), each update of $E(k)$ is influenced by $Y(k)$, making it impossible to represent using a general mathematical equation. Consequently, the stimulation of periodic signals induces unique dynamic behavior in the CCNN model. Figure 5 illustrates the waveforms and phase space plots of the CCNN neuron under square wave signal stimulation, demonstrating its complex dynamic characteristics.

Nonlinear analysis methods are subsequently utilized to investigate the dynamic behavior of the CCNN, with the equilibrium point of the model defined as follows:

$$E(k+1) = E(k) \implies$$
$$E(k)\left(1 + e^{-(U(k)-E(k))}\right) = \frac{V_E}{1 - e^{-\alpha_e}}. \qquad (8)$$

Using the Taylor series expansion of $e^x$, the above equation is simplified, resulting in the following simplified equation (9):

$$E(k)^2 - (U(k) - 2)E(k) - \frac{V_E}{1 - e^{-\alpha_e}} = 0. \qquad (9)$$

Since $V_E > 0$, $\alpha_e > 0$, and $4V_E(1 - e^{-\alpha_e}) > 0$, the discriminant $\Delta = (U(k) - 2)^2 + 4V_E(1 - e^{-\alpha_e}) > 0$. In this case, $E(k)$ can be expressed as: $E(k) = \frac{U(k)-2\pm\sqrt{(U(k)-2)^2+4V_E(1-e^{-\alpha_e})}}{2}$. Since $U(k)$ is an independent variable, equation (4) represents a two-dimensional discrete dynamic system. Using nonlinear analytical methods, it is derived that the system has two equilibrium points, which can be expressed as:

$$U(k) = \frac{e^{-k\alpha_f} \sin\left(\frac{k \cdot \pi}{2}\right) - \alpha_f e^{-k\alpha_f} \cos\left(\frac{k \cdot \pi}{2}\right)}{1 + \alpha_f^2}$$
$$E(k) = \frac{U(k) - 2 \pm \sqrt{(U(k)-2)^2 + 4V_E(1-e^{-\alpha_e})}}{2} \qquad (10)$$
$$Y(k) = \frac{1}{1 + e^{-(U(k)-E(k))}}.$$

Therefore, the CCNN neuron generates a chaotic sequence $Y(k)$ under periodic stimulation. While the processing of event signals by the CCNN is illustrated as Figure 4.

**Continuous Wavelet Transform.** In the CWT, after extensive experimentation, the Gaussian wavelet is chosen as the basis function. It is derived from the translation and scaling of the Gaussian function, as shown in equation (11):

$$\psi_{a,b}(t) = \frac{1}{\sqrt{a}} \psi\left(\frac{t-b}{a}\right),$$ (11)

where $\psi(\cdot)$ is the Gaussian function, $t$ is the input signal, and $\sigma$ is the standard deviation controlling the function's width. $\psi_{a,b}(t)$ denotes the Gaussian wavelet, with $a$ as the scale parameter and $b$ as the translation parameter determining its position.

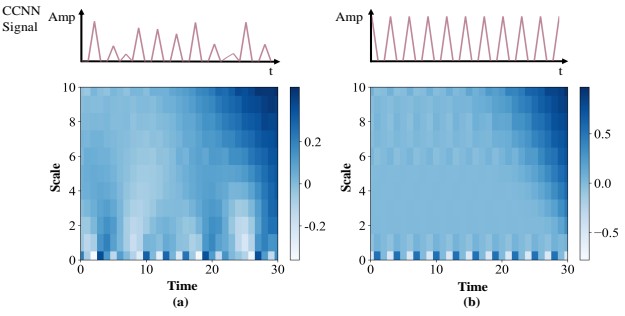

*Figure 6.* **The heatmap of the real part of the CWT matrix.** (a) The real part of the CWT matrix corresponding to the chaotic sequence. (b) The real part of the CWT matrix corresponding to the periodic sequence.

In the selection of the scale parameter for wavelet transform, we monotonically increased the scale parameter starting from 1, and ultimately chose a scale range of 10.

$$\begin{aligned} CWT(a,b) &= Y(t) \circledast \psi_{a,b}(t) \\ &= \frac{1}{\sqrt{2\pi}a\sigma} \int_0^k Y(t) e^{-\frac{(t-b)^2}{2a^2\sigma^2}} dt, \end{aligned}$$ (12)

where $Y(t)$ represents the output of the event polarity sequence processed by the CCNN, $\circledast$ represents the convolution operation.

An empirical analysis reveals that for periodic sequences, the real parts of the wavelet coefficients are predominantly negative, whereas for chaotic sequences, they exhibit the opposite trend. The results, shown in Figure 6, demonstrate that summing the real parts of all wavelet coefficients effectively distinguishes between sequences with polarity changes and those without.

$$S_{ij} = \sum_{i=1}^{10} \sum_{j=1}^{k} Re(cwt(a_i, b_j)),$$ (13)

where $S_{ij}$ represents the sum of the real parts of all elements in the coefficient matrix.

**Low-pass Filter.** The constant polarity sequence and the changing polarity sequence are processed through the CCNN and CWT, resulting in values distributed on either side of the zero point. To accurately reflect the position of the moving object, a linear low-pass filter is used to extract effective event points as the pixel points of the event frame. Its expression is given by the following equation:

$$H(f) = \begin{cases} 255, & f < 0 \\ 0, & f > 0. \end{cases}$$ (14)

We set the coordinate points less than zero to 255 and those greater than zero to 0. This processing approach not only filters out irrelevant information but also preserves the most critical dynamic changes in the event stream, resulting in a clearer and more accurate event frame $F(x,y)$:

$$F(x,y) = \sum_{i=1}^{M} \sum_{j=1}^{N} S_{ij} \cdot H(f).$$ (15)

## 4. Experiment

**Dataset.** We validate the stability and generalization of the proposed event representation method on four object classification datasets: N-MNIST (Orchard et al., 2015), N-Caltech101 (Orchard et al., 2015), N-CARS (Sironi et al., 2018), and ASL-DVS (Bi et al., 2019). Among these, N-MNIST dataset, a spiking version of the frame-based MNIST, contains 60000 training and 10000 testing samples (28×28 pixels). It was generated by capturing event streams with an ATIS sensor mounted on a motorized pan-tilt unit. N-Caltech101 dataset, derived from Caltech101, includes 8677 samples across 101 categories, with each category containing 40~800 samples (approximately 300×200 pixels). N-CARS dataset is a real-world event-based car classification dataset with 12336 cars and 11693 non-car samples, recorded using an ATIS camera capturing 100 ms events. ASL-DVS dataset comprises 100800 samples across 24 ASL letters (excluding J), with each 100 ms sample recorded using a DAVIS240c event camera in a controlled office environment.

**Experimental Details.** For each dataset, we employed a ResNet-34 architecture pre-trained on the ImageNet dataset. The data was split into training, validation, and test sets in a ratio of 3:1:1, with the random seed set to 2024. The model was trained for five epochs with a batch size of 16. During optimization, we used the cross-entropy loss function and the Adam optimizer with an initial learning rate of $1e-4$. To mitigate overfitting, we introduced a dropout layer before the fully connected layer and incorporated an early stopping mechanism during training. When the validation loss ceased

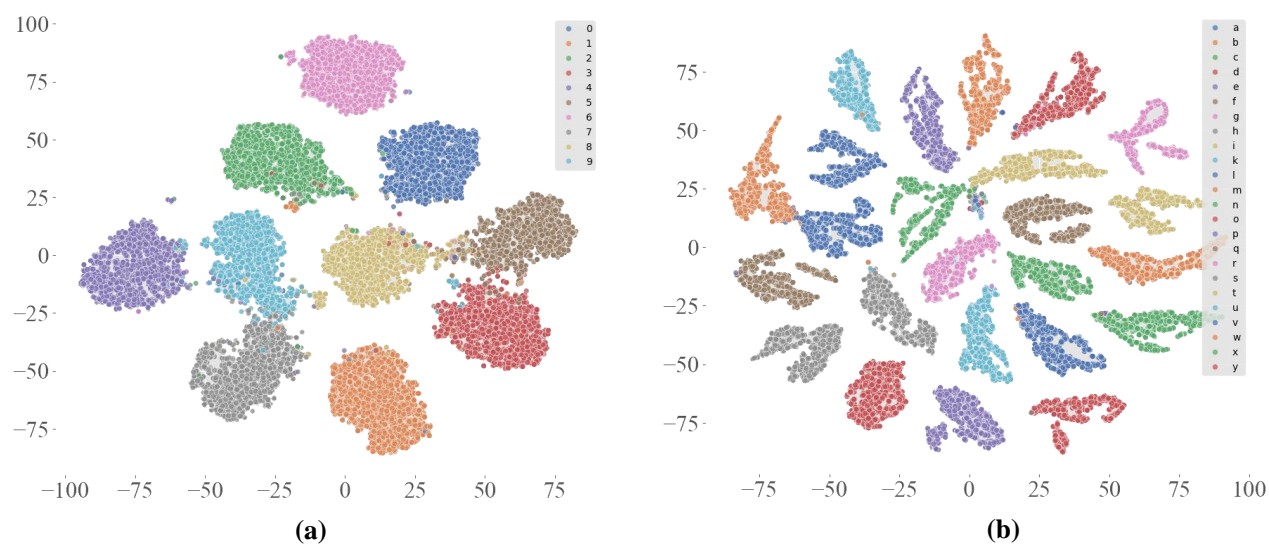

Figure 7. **Semantic distance map.** (a) Semantic 2D vector distribution of N-MNIST. (b) Semantic 2D vector distribution of ASL-DVS.

Table 1. **Classification accuracy on various datasets.** ♣ Spike-based, ♦ Voxel-based, ♥ Frame-based, ★ Ours.

| Method | N-M | N-Cal | N-Cars | ASL |
|---|---|---|---|---|
| ♣ NDA (Li et al.) | – | 78.2 | 90.1 | – |
| ♣ VPT-STS (Shen et al.) | – | 79.2 | 95.8 | – |
| ♦ GIN (Xu et al.) | 75.4 | 47.6 | 84.6 | 51.4 |
| ♦ EventNet (Sekikawa et al.) | 75.2 | 42.5 | 75.9 | 83.3 |
| ♦ RG-CNNs (Bi et al.) | 99.0 | 65.7 | 91.4 | 90.1 |
| ♦ EV-VGCNN (Deng et al.) | 99.4 | 74.8 | 95.3 | 98.3 |
| ♦ VMV-GCN (Xie et al.) | 99.5 | 77.8 | 93.2 | 98.9 |
| ♦ TORE (Baldwin et al.) | 99.4 | 79.8 | 94.5 | **99.9** |
| ♥ HATS (Sironi et al.) | 99.1 | 64.2 | 90.2 | – |
| ♥ EST (Gehrig et al.) | 99.0 | 75.3 | 91.9 | 97.9 |
| ♥ AMAE (Deng et al.) | 98.3 | 69.4 | 93.6 | 98.4 |
| ♥ M-LSTM (Cannici et al.) | 98.6 | 73.8 | 92.7 | 98.0 |
| ♥ MVF-Net (Deng et al.) | 98.1 | 68.7 | 92.7 | 97.1 |
| ♥ EvT (Sabater et al.) | 98.3 | 61.3 | 89.6 | 99.9 |
| ♥ DVS-ViT (Wang et al.) | 98.1 | 63.3 | 90.7 | 96.9 |
| ♥ TOKEN (Jiang et al.) | **99.9** | 81.6 | 95.4 | 99.9 |
| ★ **Ours** | 97.4 | **84.4** | **99.9** | 99.2 |

Table 2. **Model complexity of different methods on object classification. Here, we report average inference time on N-Cars.**

| Method | Params(M) | MACs(G) | Time(ms) |
|---|---|---|---|
| ♦ PointNet++ (Qi et al.) | 1.8 | 4.0 | 103.9 |
| ♦ RG-CNNs (Bi et al.) | 19.5 | 0.8 | – |
| ♦ EV-VGCNN (Deng et al.) | 0.8 | 0.7 | 7.1 |
| ♦ VMV-GCN (Xie et al.) | 0.8 | 1.3 | 6.3 |
| ♥ EST (Gehrig et al.) | 21.4 | 4.3 | 6.4 |
| ♥ M-LSTM (Cannici et al.) | 21.4 | 4.3 | 6.4 |
| ♥ MVF-Net (Deng et al.) | 33.6 | 5.6 | 10.1 |
| ★ **Ours** | 21.9 | 3.7 | 2.1 |

to decrease, training was terminated early to ensure robust model performance.

**Results.** As listed in Table 1, our framework outperformed all competing methods on the N-Caltech101 and N-CARS datasets and achieved competitive results on the N-MNIST and ASL-DVS datasets. On the N-CARS dataset, our framework achieved a near-perfect accuracy of 99.9%, surpassing the current best-performing TOKEN method by 4.5% and TORE by 5.4%. Even advanced approaches such as EST and MVF-Net showed inferior performance compared to our framework. This underscores our framework's ability to effectively leverage temporal and polarity information. On the N-Caltech101 dataset, our framework's accuracy exceeded HATS by 17.5% and EST by 9.05%,

demonstrating its capability to handle complex datasets with high intra-class variation. In contrast, handcrafted representations performed poorly on such datasets due to their inability to fully exploit temporal and spatial features. On the N-MNIST dataset, our framework achieved an accuracy of 97.4%, comparable to the state-of-the-art methods, EST and HATS. This indicates our framework's robustness even on datasets with lower complexity. For the ASL-DVS dataset, TORE achieved the highest accuracy of 99.9%. Our framework achieved a comparable performance with an accuracy of 99.2%, demonstrating its effectiveness in handling complex gesture recognition tasks. The semantic 2D vector distribution maps of N-MNIST and ASL-DVS are shown in Figure 7.

**Complexity Analysis.** Table 2 lists the model complexity of different methods of object classification. We evaluate the model complexity comprehensively by three metrics: the number of trainable parameters, the number of multiply–accumulate operations (MACs), and average inference time. Our framework achieves superior accuracy on N-Caltech101 while keeping the moderate model complexity(3.67G MACs), demonstrating the efficiency of our

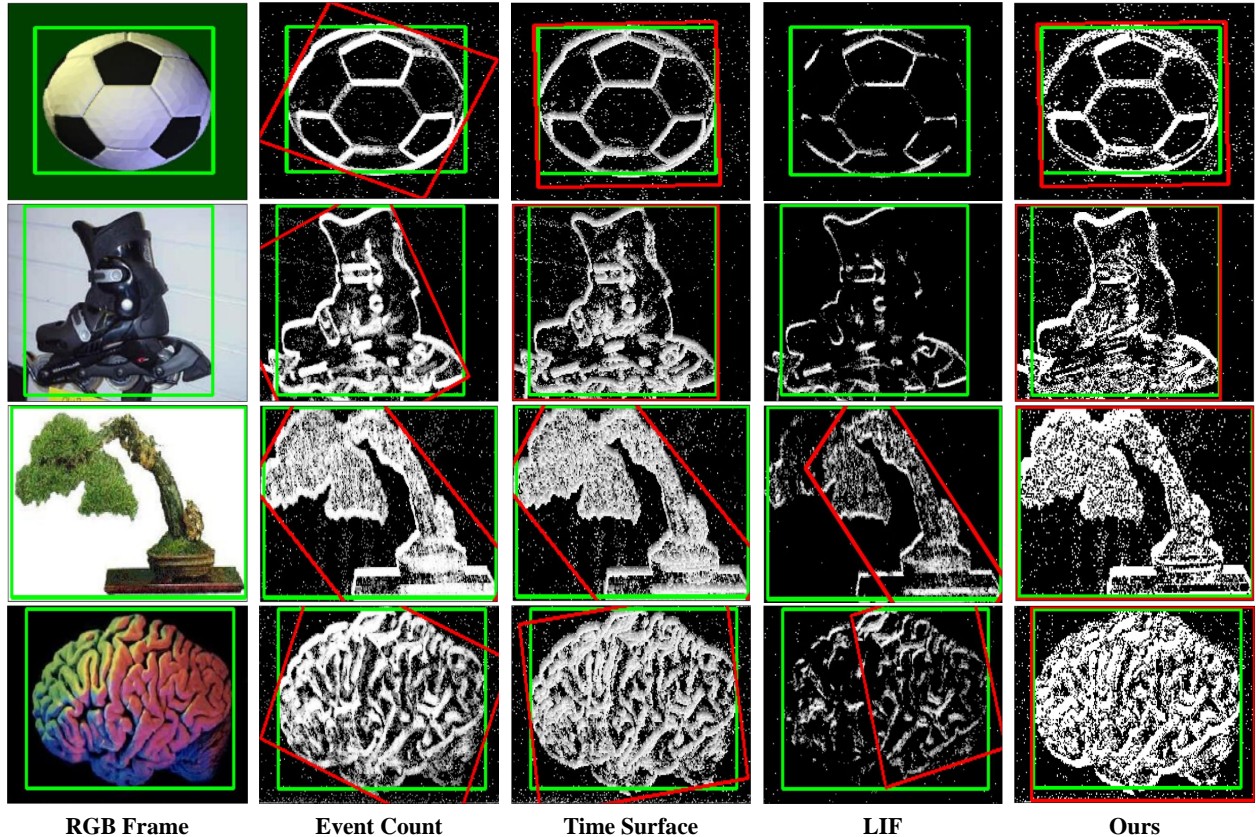

| RGB Frame | Event Count | Time Surface | LIF | Ours |

*Figure 8.* **Visualization of different event representation methods on the N-Caltech101 dataset.** Green boxes are ground truth, while red boxes are the minimum enclosing rectangle.

framework in event-based representation learning. We further measure the average inference time of our framework on N-Cars using a workstation (CPU: Intel Core i9, GPU: NVIDIA RTX 4060, RAM: 16GB). Our framework takes 2.12 ms to recognize a sample equivalent to a throughput of 472 samples per second, showing the practical potential in high-speed scenarios.

*Table 3.* **IoU of different event representations on the N-Caltech101 dataset.**

| Event Representation | IoU (30000) | IoU (50000) | IoU (70000) | IoU (100000) |
|---|---|---|---|---|
| Event Count (Miao et al.) | 0.4276 | 0.5640 | 0.5896 | 0.5937 |
| Time Surface (Miao et al.) | 0.4845 | 0.5976 | 0.6089 | 0.6198 |
| LIF (Miao et al.) | 0.2056 | 0.2162 | 0.3722 | 0.4022 |
| **Ours** | **0.4879** | **0.6087** | **0.6357** | **0.6450** |

**IoU Comparison.** The experimental results on the N-Caltech101 dataset shows the Intersection over Union (IoU) performance of our framework and other methods. The methods were evaluated under different numbers of events (30000, 50000, 70000, and 100000). As listed in Table 3, our framework shows superior performances than all other methods at each event count. Starting with an IoU of 0.4879

at 30000 events, our framework showed substantial improvement as the number of events increased, reaching an IoU of 0.6450 at 100000 events. These results suggest that our framework is more effective in extracting and utilizing spatiotemporal information from event streams, particularly as higher event counts enhance object shapes and features. The observed improvement further underscores the robustness of the model and highlights the superiority of the proposed approach. The visualization of different event representation methods on the N-Caltech101 dataset is shown in Figure 8.

## 5. Conclusion

In this work, we propose a chaotic dynamics framework inspired by the dorsal stream for event signal processing, which generates generalized and stable event representations. Then the framework is integrated with image-based algorithms for event-based object classification, achieving high accuracy across multiple datasets. Furthermore, our framework demonstrates significant efficiency in sample inference, processing 472 samples per second. In summary, we propose a method for event cameras, combining robustness with computational efficiency, and demonstrating promising application potential in real environments.

## Acknowledgement

This work is sponsored by Beijing Nova Program (2022038, 20240484703). Some experiments are supported by the Supercomputing Center of Lanzhou University. Additional support was provided in part by the Gansu Computing Center.

## Impact Statement

This paper presents work whose goal is to advance the field of Machine Learning. There are many potential societal consequences of our work, none which we feel must be specifically highlighted here.

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
