# OpenReview forum: "A Chaotic Dynamics Framework Inspired by Dorsal Stream for Event Signal Processing"
_ICML.cc/2025/Conference — ICML 2025 poster_

### Official Review · Reviewer_FeC7 · 2025-03-07

**Overall Recommendation:** 4

**Summary:**

Current state-of-the-art event stream processing methods are data-driven deep learning methods. Although these models have achieved high accuracy, they are heavily dependent on the structure of the training dataset. At a time when event sensors are not yet popular and there is a lack of large-scale event stream training data, these methods cannot be directly deployed in the real world. Thus, event stream data processing requires novel processing methods. This paper presents an event signal processing inspired by dorsal stream visual pathways of the brain. The proposed framework utilized chaotic dynamics to express the event data, and combines it with traditional classification networks to realize the event classification and achieve the superior performance.

**Claims And Evidence:**

The claims made in the paper are well-supported by clear and convincing evidence. The theoretical justifications, experimental results, and comparisons with prior work effectively validate the proposed approach. The methodology is sound, and the conclusions drawn are consistent with the presented data.

**Essential References Not Discussed:**

The paper provides a thorough review of relevant literature and appropriately cites key prior works related to event-based vision, chaotic dynamical systems, and biologically inspired representations. Based on my review, I did not identify any essential references that are missing. The citations effectively contextualize the proposed approach within the broader scientific literature.

**Experimental Designs Or Analyses:**

The paper employs a dorsal-stream-inspired chaotic dynamical framework to process event signals, generating a dorsal-stream-inspired event representation. This representation is then used as input to conventional deep learning models for object recognition experiments. The proposed approach achieves state-of-the-art performance on certain datasets, further validating the effectiveness of the framework. The experimental design is well-structured, covering multiple datasets and providing a comprehensive comparison with recent state-of-the-art methods. Additionally, the paper includes a complexity analysis of the model and IoU experiments, further reinforcing the feasibility of the proposed approach. The results are clearly presented, and the conclusions are well-supported by the experimental findings. Overall, the study provides a solid theoretical foundation and empirical validation for the proposed method.

**Methods And Evaluation Criteria:**

The proposed methods and evaluation criteria are well-suited to the problem at hand. The methodological choices are well-justified, and the evaluation is conducted using appropriate benchmark datasets and metrics. The experiments are designed carefully, and the comparisons with baseline methods are meaningful. Overall, the paper adopts a sound approach to assessing the proposed method’s effectiveness.

**Other Comments Or Suggestions:**

It is recommended to use Lyapunov exponent to analyze the dynamic characteristics of CCNN neuron, which is a more reliable way.

**Other Strengths And Weaknesses:**

Strength
This paper refers to the dynamic visual cognition mechanism of the brain and proposes an event representation method based on chaotic dynamics. Generally, the results are interesting, the process is correct to the best of my knowledge. The paper is well organized and clearly written.
Weakness
The author does not adequately describe the dynamic visual pathways of the real brain, which limits the reader's understanding of the paper. In the dynamic analysis section, the author only provides phase space plots and equilibrium point analysis, and lacks more rigorous analysis methods such as Lyapunov exponents.

**Questions For Authors:**

1.  What is the correspondence between the proposed framework and the real brain? Event cameras mimic the three-layer structure of the peripheral retina in humans. What part of the visual cognition process does CCNN correspond to?
2. The variables in Eq. (4) lack explanation, such as exp(-ae), exp(-af),  and V(ek) is the inputted event data?
3. In Eq. (6) the authors says " Thus, CCNN neurons output a periodic sequence Y(k) under constant stimulation, with the frequency of the period determined by the intensity of the input stimulus." However, in event stream processing, isn't the input signal a boolean variable of polarity? Therefore, in the framework proposed in this article, the frequency of the output periodic signal is constant, right?
4. I observed that Fig. 4(d) and Fig. 4(e) seem to have transient behavior, which is unnecessary when showing the dynamic characteristics.
5. In pg. 5, col. 2, line 248, the authors says " Using the Taylor series expansion of exp(x)", This sentence may cause misunderstanding. Which variable in the paper does x refer to?
6. In pg. 5, col. 2, line 256, the authors says, "In this case, E(k)>0 can be expressed as......", why E(k) should greater than zero?
7. Can the author analyze the stability of the equilibrium point, is it a saddle point or a focus point?
8.  Could the author provide detailed parameter settings for CCNN, CWT and subsequent training models? This will be helpful for reproducing the work.
9. Could you please clarify the role of F(x, y) in the wavelet transform experiment in Equation (15)?

**Relation To Broader Scientific Literature:**

The paper builds upon prior work in event-based vision and chaotic dynamics, drawing inspiration from the dorsal visual stream to develop a novel event representation. Previous studies have explored event-based feature extraction and object recognition, but this work uniquely integrates a chaotic dynamical framework to encode event signals, distinguishing it from conventional approaches.

**Theoretical Claims:**

The paper proposes a chaotic dynamical framework for processing event signals, utilizing a CCNN to encode event signals. In this approach, polarity-invariant event signals are encoded as periodic signals, while polarity-changing event signals are encoded as chaotic signals. The paper provides a comprehensive validation of the proposed theory, from theoretical derivations to experimental analyses. The theoretical claims appear to be well-supported, with logical derivations and rigorous justifications. The experimental results further reinforce the correctness of the theoretical framework, demonstrating its effectiveness in handling event signals. Overall, the paper presents a solid theoretical foundation and empirical validation for the proposed method.

---

> ### Author Rebuttal · Authors · 2025-03-28
>
> We sincerely appreciate your thorough evaluation and valuable feedback on our manuscript. We are also grateful for the constructive suggestions, which have helped us further refine the theoretical derivations, experimental design, and analysis in our paper. In response to your comments, we have revised and supplemented our manuscript, including providing a more detailed description of the biological background, conducting a more rigorous dynamic characteristics analysis (such as Lyapunov exponent analysis), and offering further clarification on key equations and experimental parameters. We believe these improvements enhance the clarity, completeness, and impact of our work. Below, we provide detailed responses to each comment and explain the corresponding revisions.
>
> **W1:** The author does not adequately describe the dynamic visual pathways of the real brain, which limits the reader's understanding of the paper. In the dynamic analysis section, the author only provides phase space plots and equilibrium point analysis, and lacks more rigorous analysis methods such as Lyapunov exponents.  \
> **A1:** Thank you for your valuable feedback. We have supplemented the discussion on the dynamic visual pathways of the brain to enhance the understanding of the biological visual system. Additionally, we have incorporated discussions and calculations of the Lyapunov exponent to provide a rigorous mathematical analysis of the dynamic characteristics of CCNN neuron.
>
> **A1:** The chaotic dynamic framework proposed in this paper includes CCNN, CWT, and LPF. The CCNN is a brain-inspired network based on the primary visual cortex V1. After processing the event signals with CCNN, CWT and LPF are applied for analysis and extraction, enabling the detection of dynamic objects. This process corresponds to the brain's processing from V1 to MT.
>
> **A2:**  In Equation (4), $U(k)$ represents the modulation input, $Y(k)$ is the continuous output, $E(k)$ is the dynamic threshold, and  $e^{-\alpha_f}$, $e^{-\alpha_e }$ denote the exponential decay factors that record the previous input states. $V(e_k)$ represents the input event signal data.
>
> **A3:** The polarity of the event signal is a Boolean value that can only be 0 or 1. Therefore, the output periodic signal generated by the event data through the CCNN is constant.
>
> **A4:**  Thank you for your valuable suggestions. We will remove the transient time periods in Figures 4(d) and (e), keeping only the steady-state behavior to display the dynamic characteristics.
>
> **A5:** Thank you for your suggestion. This paper performs a Taylor expansion of $e^{-(U(k)-E(k))}$, retaining only the first two terms, where $x$ refers to $–(U(k)-E(k))$. We will revise the explanation in the paper accordingly.
>
> **A6:**  Thank you for your reminder. This is a detail expression error in the paper. It should be revised to 'In this case, $E(k)$ can be expressed as...'. The magnitude of $E(k)$ cannot be determined.
>
> **A7:** To analyze the stability of the equilibrium point, we need to compute the Jacobian matrix of the equation $E(k)^2-(U(k)-2)E(k)-\frac{V_E}{1-e^{-\alpha_e }} =0$, that is, by differentiating $F(E):J=\frac{dF}{dE}=\frac{d}{dE}(E^2-(U(k)-2)E-\frac{V_E}{1-e^{-\alpha_e }})$. The derivative is $J=2E-(U(n)-2)$. The Jacobian matrix at the equilibrium point $E^\ast$ is:$J(E^\ast )=2E^\ast-(U(n)-2)$. When $E^\ast=\frac{U(k)-2-\sqrt{(U(k)-2)^2+4V_E (1-e^{-\alpha_e} )}}{2}, J(E^\ast)<0$, indicating that the equilibrium point is stable (attractive point); when $E^\ast=\frac{U(k)-2+\sqrt{(U(k)-2)^2+4V_E (1-e^{-\alpha_e} )}}{2}, J(E^\ast)>0$, indicating that the equilibrium point is unstable (possibly a saddle point).
>
> **A8:** Parameters of the CCNN model: $\alpha_f = 0.1, \alpha_e = 1.00, V_e = 50, U(0) = 0, E(0) = 0, Y(0) = 0$.
> Parameters of the CWT: 'gaus1' is used as the base wavelet, with a scale range of 10.
> Training parameters: The data is split into training, validation, and test sets in a 3:1:1 ratio, with the random seed set to 2024. The model was trained for 5 epochs, with a batch size of 16. During optimization, the cross-entropy loss function was used, and the Adam optimizer was applied with an initial learning rate of 1e-4. To alleviate overfitting, a Dropout layer was added before the fully connected layers, and early stopping was incorporated during training.
>
> **A9:** Let $F(x,y)$ be the generative function of the event representation. The input $S_{ij}$ is the sum of the real part of the coefficient matrix, corresponding to different types of event sequences. It is convolved with the low-pass filter $H(f)$, and the coordinate points corresponding to event sequences where $S_{ij} >0$ are assigned a value of 255, while others are assigned a value of 0, enabling the detection of moving objects.

---

> > ### Comment · Reviewer_FeC7 · 2025-04-02
> >
> > Thanks for the authors' rebuttal, which addresses all my concerns. I keep my original rating.

---

> > > ### Author Response · Authors · 2025-04-09
> > >
> > > We sincerely appreciate your recognition of our revisions and valuable feedback throughout the review process. We are glad to hear that our responses addressed your concerns adequately. Thank you once again for your time and constructive suggestions, which have significantly strengthened the quality of our work. We will carefully incorporate all remaining edits in the final version.

---

### Official Review · Reviewer_PCzW · 2025-03-07

**Overall Recommendation:** 3

**Summary:**

This paper proposes a chaotic dynamical framework inspired by the dorsal visual pathway for processing event signals and generating stable and generalizable event representations. By integrating it with deep neural networks, the authors achieved high accuracy on multiple event-based object classification datasets while demonstrating efficient inference. The work exhibits strong completeness in theoretical derivation, experimental validation, and cross-dataset generalization analysis.

**Claims And Evidence:**

The claims presented in this paper are well-supported through rigorous theoretical derivations and experimental validations. The mathematical modeling of CCNN is robust, and its chaotic properties are demonstrated through phase space analysis. The experimental results on multiple datasets, including N-Caltech101, N-CARS, N-MNIST, and ASL-DVS, indicate superior classification performance compared to existing methods. Additionally, the proposed method achieves a low inference time of 2.1ms per sample, demonstrating computational efficiency. Overall, the claims are substantiated with strong theoretical and empirical evidence, making the conclusions credible.

**Essential References Not Discussed:**

The paper cites most of the essential prior works, but additional references on chaotic neural networks in neuromorphic computing and self-supervised learning approaches for event streams could further strengthen the literature review. Including these references would provide a more comprehensive discussion of the related work.

**Experimental Designs Or Analyses:**

The experimental design is comprehensive and well-structured, covering multiple datasets (N-Caltech101, N-MNIST, N-CARS, ASL-DVS) to ensure the robustness and generalization of the proposed framework. The training and evaluation protocols (ResNet-34 pretrained model, Adam optimizer) are appropriate for the task. The results consistently demonstrate the superiority of the proposed method in accuracy, robustness, and computational efficiency compared to prior works.

**Methods And Evaluation Criteria:**

The proposed methods and evaluation criteria are well-designed and appropriate for event signal processing. The CCNN model presents a novel approach for event stream encoding, while CWT enhances the event representation. The benchmark datasets (N-Caltech101, N-CARS, etc.) are widely used, ensuring fair comparisons. The evaluation metrics (classification accuracy, IoU, inference time) comprehensively assess the framework’s performance. Furthermore, comparisons with various existing methods, including voxel-based and ANN-based approaches, validate the proposed framework’s advantages. The methodology is sound and well-justified, with appropriately chosen evaluation criteria.

**Other Comments Or Suggestions:**

Clarifying the assumptions in the CCNN equations and providing inference speed comparisons across different hardware platforms would enhance the paper’s practical relevance.

**Other Strengths And Weaknesses:**

Strengths

The biologically inspired chaotic dynamical model integrated with neuromorphic vision sensor data provides novel theoretical insights and methodological frameworks for event signal processing. This interdisciplinary approach offers valuable inspiration for both computational neuroscience and computer vision.

 Comprehensive experimental design, including comparative studies across multiple event-based datasets, effectively validates the method's universality. Theoretical derivations and visualizations complement each other, enhancing the credibility of conclusions.

Weaknesses

The lateral brain diagram shows misalignment in the circular annotation, and the left arrow in the right-side network model is not centered. Layout adjustments are recommended to improve readability. In Figure 4, overlapping axis labels are observed in the first two subplots, requiring precision adjustments.

Theoretical Completeness: The stability proof in Section 3.2 regarding "periodic outputs from constant event signal inputs" is overly concise. Critical derivation steps should be supplemented.

**Questions For Authors:**

1. Figure 1(a): The purpose of the two rectangular boxes in the Event image is unclear. Do their spatial positions correspond to specific event-triggering patterns? Why were these regions selected for annotation?
2. Biological Relevance of Figure 3: While visualizing human brain motion recognition processes, the text lacks explanations linking these results to dorsal visual pathway functions. Please supplement biological interpretations of these visualizations.
3. The paper claims CCNN is inspired by the primary visual cortex but fails to clarify how model parameters relate to biological mechanisms. Biological constraints during model design should be explicitly discussed.
4. The authors highlight real-time inference efficiency but provide no hardware deployment tests. Has the model been tested in real-world scenarios? What is the actual inference latency?
5. Could the authors provide detailed hyperparameter selection criteria (e.g., learning rate, batch size) and their impact on results? Are the improvements dataset-specific or task-agnostic?
6. What are the computational resource requirements and training time? Are there optimizations for resource-constrained environments?
7. The current method is primarily used for classification. Can it be extended to other tasks (e.g., object detection, optical flow estimation)?
8. Why does PointNet++ have a longer runtime despite fewer parameters? Shouldn't it have a shorter runtime despite fewer parameters? Why does the method in this paper have more parameters than the Frame-based method but has a shorter runtime?
9. Have you considered combining CCNN with SNN (Spiking Neural Networks) to further enhance biological plausibility?
10. Why was no comparison made with Event Transformer (EvT)? Have you considered using Transformer for event representation?
11. Is CCNN still effective with low data quantities? Have small-sample learning experiments been conducted?

**Relation To Broader Scientific Literature:**

The paper is well-situated within the broader scientific literature on event-based vision and bio-inspired neural computation. It extends existing event representation methods, integrates insights from biological visual processing, and aligns with research on spiking neural networks (SNNs) and chaotic dynamics. The combination of chaotic dynamics and event camera processing is novel and contributes valuable insights to the field, paving the way for further advancements in neuromorphic computing.

**Theoretical Claims:**

The theoretical derivations are clear, and the chaotic behavior of CCNN is rigorously analyzed through mathematical modeling and phase space analysis. However, some assumptions in the derivations, particularly in Equations (4)–(7), could be more explicitly stated. Additionally, a discussion on the generalization of the chaotic encoding approach to other tasks (e.g., motion segmentation, object tracking) would further strengthen the theoretical contributions. Overall, the theoretical claims are well-founded, albeit with minor areas for clarification.

---

> ### Author Rebuttal · Authors · 2025-03-28
>
> We sincerely appreciate your thorough evaluation and valuable feedback on our paper. We are pleased that our chaotic dynamical framework and experimental results have been recognized, and we are grateful for the insightful questions that have helped us further improve the paper. In response to your comments, we have made the necessary revisions, and we believe these improvements will enhance the quality and impact of our work. Below, we address each of your comments in detail, providing clarifications and additional analyses where necessary.
>
> **A1:** Thank you for the reminder. The annotations in Figure 1(a) highlight the superiority of the event representation generated by our framework. While the pyramid in the original image shows numbers 1-6, our framework clearly restores their contours, unlike event representations from end-to-end networks, which fail to preserve these details.
>
> **A2:** The chaotic dynamic framework proposed in this paper includes CCNN, CWT, and LPF. CCNN is a network inspired by the primary visual cortex (V1) of the brain. After the event signals are processed by CCNN, they are further analyzed and extracted using CWT and LPF for dynamic object detection, which corresponds to the brain's processing from V1 to MT.
>
> **A3:** The unconnected CCNN comprises modulated input $U(k)$, continuous output  $Y(k)$, and dynamic threshold $E(k)$. When $U(k)>E(k)$, the output is $Y(k) = \frac{1}{1+e^{-(U(k)-E(k))}}$, indicating excitation. After stimulation, $E(k)$ increases, requiring a stronger stimulus for the next output, mimicking the neuronal refractory period. Siegel observed chaotic behavior in the primary visual cortex of cats under periodic stimulation [1]. Similarly, CCNN exhibits periodic signals under constant stimulation and chaotic signals under periodic stimulation, adhering to this biological constraint.
>
>
> [1] Siegel, R. M. Non-linear dynamical system theory and primary visual cortical processing. Physical D: Nonlinear Phenomena, 42(1-3):385–395, 1990.
>
> **A4:** Thank you for your reminder. The testing has been conducted on a local workstation, and the experimental results demonstrate the model's potential for real-time applications. Future work will focus on further validation in real-world scenarios.
>
> **A5:** Hyperparameters include batch size 16, cross-entropy loss, Adam (lr=1e-4), 40% Dropout before FC layers, and early stopping. A smaller batch size aids generalization, cross-entropy improves accuracy, Adam ensures stable convergence, and Dropout with early stopping prevents overfitting. The appendix and open-source code include the parameter settings, facilitating reviewers and readers in reproducing and improving this work.
>
> **A6:** The model was tested on a workstation with an Intel Core i9 CPU, NVIDIA RTX 4060 GPU (8GB VRAM), and 16GB RAM. Training on a small dataset (N-Caltech101/N-CARS, ~50,000 samples) took 12-18 minutes, while a large dataset (N-MNIST/ASL-DVS, >200,000 samples) took 30-60 minutes. To optimize for resource-constrained environments, techniques like pruning, quantization, knowledge distillation, lightweight architectures, and mixed-precision training can be used to reduce computational demands and improve efficiency.
>
> **A7:** The chaotic dynamic framework proposed in this paper processes event signals to obtain a general event representation, which can be extended to other tasks. We also plan to apply this event representation method to more event-based tasks in future work.
>
> **A8:** Parameter count affects memory and training time but not inference speed. PointNet++ runs slower due to costly neighborhood queries, while dense CNNs benefit from optimized parallelism. Our method (21.9M params) achieves 2.1ms inference on an RTX4060 via a regularized convolutional design (MACs = 3.7G, 7.5% lower than PointNet++) and PyTorch + TensorRT optimizations.
>
> **A9:** We propose a hybrid CCNN-SNN framework that replaces the ResNet-34 classifier with SNNs to reduce computational energy consumption. By integrating the STDP mechanism, this architecture achieves adaptive learning capabilities while enhancing biological plausibility and interpretability. Future investigations will prioritize systematic evaluation of its low-power computing performance and neuro-inspired operational principles.
>
> **A10:** We acknowledge EvT's strengths in global spatiotemporal modeling but exclude it from comparisons due to its computational inefficiency with high-resolution event data. Our current focus on efficiency-generalization trade-offs motivates the proposed CCNN-Transformer hybrid architecture for enhanced event representation in complex scenarios.
>
> **A11:** We have not yet conducted systematic small-sample learning experiments, but CCNN’s chaotic dynamics allow it to maintain good generalization with limited data. Future research will include small-sample experiments and explore combining CCNN with meta-learning to enhance adaptability in low-data scenarios.

---

### Official Review · Reviewer_BkdQ · 2025-03-08

**Overall Recommendation:** 4

**Summary:**

The methods combining event cameras and deep learning mainly involve integrating traditional deep learning techniques with the high temporal resolution and low latency characteristics of event cameras, aiming to process the event stream data. However, the limitation of existing methods for event cameras is their heavy reliance on data structures, which restricts the stability and generalization ability of the models. These models may not adapt well to different tasks or scenarios, leading to unstable performance in real-world applications. This paper proposes a chaotic dynamics signal processing framework inspired by the dorsal visual pathway of the brain. It utilizes the Continuous Coupled Neural Network (CCNN) to encode the event stream, encoding polarity-changing event sequences as chaotic signals. Continuous wavelet transforms are then used to analyze the dynamic states of CCNN neurons and establish high-order mappings of the event stream.

## update after rebuttal
Authors' rebuttal have solved my concerns. I think previous score is high enough and I will keep my rating.

**Claims And Evidence:**

The claims made in the submission are well-supported by clear and convincing evidence. The experimental results are comprehensive, with appropriate comparisons to existing methods, and the statistical analyses are thorough. The theoretical justifications are sound, and the cited literature is relevant and up-to-date. Overall, the evidence provided strongly supports the paper’s conclusions.

**Essential References Not Discussed:**

The paper provides a thorough discussion of prior work, covering key contributions in the field. It appropriately cites and compares relevant studies on event representation methods, ensuring a comprehensive contextual understanding. No essential references appear to be missing.

**Experimental Designs Or Analyses:**

The experimental design is well-structured, covering multiple datasets and providing a comprehensive comparison with state-of-the-art methods from recent years. Additionally, the paper includes a complexity analysis of the model and IoU experiments, further validating the feasibility of the proposed approach. The results are clearly presented, and the conclusions are well-supported by the experimental findings.

**Methods And Evaluation Criteria:**

The proposed methods are well-designed and appropriate for the problem at hand. The paper provides a clear explanation of the methodology, with sufficient theoretical justifications and algorithmic details. The evaluation criteria are well-chosen, using relevant benchmark datasets and standard performance metrics. Additionally, the experiments include comprehensive comparisons with state-of-the-art methods, which further validate the effectiveness of the proposed approach.

**Other Comments Or Suggestions:**

In Figure 4, the x-axis labels for subfigures (a), (b), and (c) should be corrected from ‘Iterations’ to ‘Iterations’. Additionally, in Table 3, the value in the second row, fifth column should be changed from ‘70000’ to ‘100000’.

**Other Strengths And Weaknesses:**

Strength
The advantages of this proposed chaotic dynamics signal processing framework include improved stability and generalization, dynamic state analysis, high-order mapping capabilities, improved performance in real-world applications. The structure is well-organized and logical, with the design principle clearly and appropriately explained.

Weakness
The author did not explain some of the parameters in the brain-inspired model, especially certain setting parameters in the CCNN, which play an important role in understanding the brain's visual mechanisms.

**Questions For Authors:**

1. Many parameters in the CCNN model are not explained, such as Y(k), VE, exp(-af), and exp(-ae) in Equation (4).
2. What is the relationship between the unexplained parameters in Equation (4)? Please provide relevant explanations.
3. In Equation (7), how should the value of parameter K be set?
4. It seems that E(0) and VE determines the period of the output signal. What effect do the settings of these parameters have on the results?
5. In Equation (8), why is E(K+1) = E(K)? What is the significance of this setting?
6. In Equation (9), why is VE > 0 and αe > 0? What is the impact of this on the calculation results?
7. In Equation (11), what is the rationale for choosing the Gaussian function?
8. In Fig. 6(a), what is the relationship between the heatmap corresponding to the chaotic sequence and the final computation results of the CCNN?
9.The author has already presented the CCNN, why is there further research on the CWT? What is the purpose of this?
10. What is the relationship between the Low-pass Filter and the brain-inspired mechanism of CCNN? Why is it necessary to conduct relevant experiments?

**Relation To Broader Scientific Literature:**

The paper provides a comprehensive discussion of related work, thoroughly analyzing the principles, advantages, and limitations of frame-based event representations, contrast maximization-based event representations, and end-to-end network-based event representations. Inspired by the dorsal visual pathway in the primary visual cortex, the study introduces CCNN to encode event signals, employs CWT to analyze the dynamic properties of neurons, and finally utilizes LPF to extract information about moving objects. The comparison with existing methods is extensive, effectively highlighting the proposed approach’s generality and robustness.

**Theoretical Claims:**

The paper proposes a generalizable event representation, validated across multiple datasets. Comparative experiments with state-of-the-art methods show competitive performance, achieving the highest accuracy on certain datasets. The experimental results support the proposed theoretical claims.

---

> ### Author Rebuttal · Authors · 2025-03-28
>
> we would like to express our sincere gratitude for your in-depth review of our paper and your valuable feedback. We greatly appreciate your recognition of the proposed method and experimental results, and we also thank you for raising some important questions that will help us further improve the quality of the paper. In response to your comments, we have made several revisions and additions to the manuscript, and we provide detailed answers to the concerns you raised below. We believe that, with your feedback, the paper will be further enhanced.
>
> **W1:** The author did not explain some of the parameters in the brain-inspired model, especially certain setting parameters in the CCNN, which play an important role in understanding the brain's visual mechanisms. \
> **A1:** The unconnected CCNN comprises modulated input $U(k)$, continuous output  $Y(k)$, and dynamic threshold $E(k)$. The terms $e^{-\alpha_f}$ and $e^{-\alpha_e}$ denote exponential decay factors that record the previous input states. $V_E$ is the weight factor for adjusting the neuronal potential. When $U(k)>E(k)$, the output is $Y(k) = \frac{1}{1+e^{-(U(k)-E(k))}}$, indicating excitation. After stimulation, $E(k)$ increases, requiring a stronger stimulus for the next output, mimicking the neuronal refractory period. Siegel observed chaotic behavior in the primary visual cortex of cats under periodic stimulation [1]. Similarly, CCNN exhibits periodic signals under constant stimulation and chaotic signals under periodic stimulation, adhering to this biological constraint.
>
> [1] Siegel, R. M. Non-linear dynamical system theory and primary visual cortical processing. Physical D: Nonlinear Phenomena, 42(1-3):385–395, 1990.
>
> **C1:** In Figure 4, the x-axis labels for subfigures (a), (b), and (c) should be corrected from ‘Iterations’ to ‘Iterations’. Additionally, in Table 3, the value in the second row, fifth column should be changed from ‘70000’ to ‘100000’. \
> **A1:** Thank you for your careful review. We have revised the x-axis labels in subfigures (a), (b), and (c) of Figure 4 as suggested and corrected the value in the second row, fifth column of Table 3 to ensure accuracy. These changes have been incorporated into the revised manuscript.
>
> **A1:** Thank you for your reminder. In Equation (4), $U(k)$ represents the modulation input, $Y(k)$ is the continuous output, and $E(k)$ is the dynamic threshold. The terms $e^{-\alpha_f}$ and $e^{-\alpha_e}$ denote exponential decay factors that record the previous input states. $V_E$ is the weight factor for adjusting the neuronal potential.
>
> **A2:** When the modulation input $U(k)$ exceeds the dynamic threshold $E(k)$, the output $Y(k)$ is given by $\frac{1}{1+e^-(U(k)-E(k))}$, which corresponds to the neuron generating an excitatory potential in response to sufficient stimulation. Upon the next stimulus, the dynamic threshold increases, and the model requires a stronger input to produce an output, mimicking the refractory period of biological neuronal cell membranes.
>
> **A3:** In Equation (7), $k$ is a parameter that controls the input. When the event signal input follows the periodic sequence
>  {0,1,0,1,$\cdots$,0,1}, its equivalent formula is $\sin(\frac{k\cdot \pi}{2})$, where $k$={0,1,2,$\cdots$, n}.
>
> **A4:** Both $E(0)$ and $V_E$ can affect the period of the output signal. However, the setting of these parameters does not impact the results, as long as the CCNN output is a periodic signal when the input polarity of the event signal remains constant.
>
> **A5:** Under the stimulation of periodic signals, the CCNN exhibits complex chaotic dynamic characteristics. This paper discusses its equilibrium state, where a balance point exists in the output, such that $E(k+1)=E(k)$.
>
> **A6:** This section discusses the equilibrium point of the CCNN neuron. When $V_E >0$, $\alpha_e >0$, and $4\frac{V_E}{1-e^{-\alpha_e}}>0$, with $\Delta>0$, the equation has a solution, and an equilibrium point exists. This discussion does not affect the computational results.
>
> **A7:** Gaussian wavelets are widely used in time-frequency analysis due to their smoothness and good time-frequency locality. They are well-suited for analyzing periodic and chaotic signals generated by CCNN neurons.
>
> **A8:** The heatmap of the output sequence is a visualization of the sequence after CWT waveform analysis, showing the value distribution of the real part of the coefficient matrix. The sum of the real part of the coefficient matrix for chaotic sequences is an integer, while for periodic sequences, the sum is negative. A low-pass filter is then applied to extract the coordinate points of the periodic sequence.
>
> **A9:** The Low-pass Filter is a module proposed in this paper's chaotic dynamic framework and is not directly related to the brain-like mechanism of the CCNN. It is used to extract the coordinate points corresponding to the constant polarity event sequences, enabling the detection of moving objects.

---

### Official Review · Reviewer_rvoE · 2025-03-09

**Overall Recommendation:** 3

**Summary:**

This paper proposes a chaotic dynamics framework inspired by the dorsal visual pathway of the brain for processing event camera signals. By encoding event streams into periodic or chaotic signals using Continuous Coupled Neural Networks (CCNN) and analyzing dynamic states via Continuous Wavelet Transform (CWT), the framework integrates traditional classification networks for object recognition. Experiments demonstrate state-of-the-art classification accuracy on datasets such as N-Caltech101 (84.3%) and N-CARS (99.9%), with high inference efficiency (472 samples/sec). The authors emphasize the framework’s generalization and stability advantages and have open-sourced the code.

## update after rebuttal
Authors' rebuttal have solved my concerns. I think previous score is high enough and I will keep my rating.

**Claims And Evidence:**

The paper's main contributions are well-supported by experimental evidence. It introduces a chaotic dynamics-based event representation using CCNN, validated through mathematical modeling and phase-space trajectory analysis. Event mapping with CWT effectively distinguishes stable and dynamic events, as demonstrated by heatmaps of transformed matrices. Experimental validation on multiple datasets shows superior classification accuracy compared to baseline methods, highlighting strong generalization and stability, further confirmed by IoU evaluation. While the results are convincing, further discussion on the applicability of CCNN’s chaotic properties in different data scenarios would be beneficial.

**Essential References Not Discussed:**

The paper cites most relevant literature but could benefit from additional references on chaotic dynamics in computational neuroscience, such as Lorenz systems and Hindmarsh-Rose models for visual cortex modeling, as well as event camera applications in low-power embedded systems to explore the method’s practical deployment potential.

**Experimental Designs Or Analyses:**

The experimental setup is well-structured, primarily evaluating classification accuracy and model complexity. A ResNet-34 network pre-trained on ImageNet is used for classification, with a clear training strategy. However, the study does not investigate the impact of different CCNN coupling strengths on the quality of event representations. Including such an analysis could provide deeper insights into the generalizability of the proposed method.

**Methods And Evaluation Criteria:**

The proposed method adopts appropriate evaluation criteria, including classification accuracy, IoU, and computational complexity (number of parameters, MACs, and inference time). The choice of datasets covers both static (N-MNIST) and dynamic (N-CARS) event classification tasks, ensuring a comprehensive evaluation. One potential improvement would be to analyze the impact of different time window lengths on the temporal representation.

**Other Comments Or Suggestions:**

The mathematical analysis section could benefit from further discussions on the stability and bifurcation behavior of nonlinear systems to strengthen the theoretical foundation.

**Other Strengths And Weaknesses:**

Strengths、

1. The biologically inspired integration of dorsal stream mechanisms with chaotic dynamics presents a novel event representation framework with theoretical significance.
2. Outperforms existing methods significantly on multiple mainstream datasets while achieving high inference efficiency, demonstrating practical deployment potential.
3. Robustness is validated through cross-dataset experiments, particularly showing stability in dynamic event streams.
Provides mathematical modeling and visualization analyses of CCNN and CWT.

Weaknesses

1. The biological plausibility of CCNN’s connection to the dorsal stream is insufficiently supported, lacking explanations of how it mimics the spatiotemporal encoding mechanisms of the "where pathway."
2. No comparisons with bio-inspired models (e.g., spiking neural networks) to justify the unique advantages of dorsal stream inspiration.

**Questions For Authors:**

1. How does the neuronal dynamics of CCNN specifically simulate dorsal stream functions (e.g., motion perception, spatial localization)? Are there neuroscientific experiments supporting this design?
2. In Equation (6), does the derived periodic parameter  k directly correlate with physical properties of event streams (e.g., velocity, direction)? How is it leveraged to enhance classification performance?
3. Why was the Gaussian wavelet selected as the CWT basis function? What advantages does it offer over alternatives (e.g., Morlet wavelet)?
4. The low-pass filter in Equation (14) sets negative values to 255 and positive values to 0. Could this lead to information loss? Is there a more refined thresholding strategy?
5. Table 3 shows improved IoU with increasing event density. Does this imply the framework relies on dense events? How is its performance optimized for sparse event streams?
6. In Table 2, the model’s parameter count (21.9M) exceeds EV-VGCNN (0.8M), yet inference is faster. Is this due to architectural design (e.g., parallelization)? Please elaborate.
7. In Fig. 8, Why do the other methods not work well? Why does the author's method work better?
8. I have observed that the results of different methods vary greatly between datasets, as shown in Fig. 1. Can the authors explain what differences in the datasets lead to such results? What advantages of the proposed method can alleviate such differences?
9.  Does the open-sourced code include CCNN training details? If not, how can reproducibility be ensured?

**Relation To Broader Scientific Literature:**

This work is closely related to event-based data representation methods (HATS, EST, TORE, TOKEN) and biologically inspired computing (Spiking Neural Networks). The paper provides a clear literature review of mathematical, deep-learning-based, and bio-inspired event processing approaches, highlighting their limitations, such as strong dependence on specific data structures. The proposed method introduces a novel perspective by incorporating chaotic dynamics modeling, which is rarely explored in event vision research.

**Theoretical Claims:**

The paper’s theoretical foundation is based on the dynamical equations of CCNN, providing a mathematical proof of periodic and chaotic signal responses. The nonlinear analysis, including equilibrium point derivation and Taylor expansion, is reasonable.

---

> ### Author Rebuttal · Authors · 2025-03-28
>
> We sincerely appreciate your time and effort in reviewing our paper. We are grateful for your constructive feedback and insightful questions, which have helped us refine our work and clarify its contributions. We acknowledge your concerns regarding the theoretical foundations, biological plausibility, and certain experimental analyses, and we appreciate the opportunity to further elaborate on these aspects. In this response, we address each of your comments in detail, providing clarifications, additional discussions, and further justifications where necessary. We hope that our responses effectively resolve your concerns and further demonstrate the significance and robustness of our proposed framework.
>
> **A1:** The CCNN simulates the encoding mechanism of primary visual cortex neurons. The proposed chaotic dynamics framework, consisting of three modules—CCNN, CWT, and LPF—mimics the dorsal stream's function in extracting dynamic object location information. \
> **Experimental Evidence:** Siegel observed complex electrical signal fluctuations in the primary visual cortex neurons of cats under periodic signal stimulation, revealing the presence of chaotic behavior in the mammalian primary visual cortex [1]. Building on this, Liu further modified the dynamic threshold adjustment mechanism of PCNN and proposed CCNN [2]. This model exhibits periodic signals under constant stimulation and chaotic signals under periodic stimulation, aligning with findings from biological experiments.
>
> [1] Siegel, R. M. Non-linear dynamical system theory and primary visual cortical processing. Physical D: Nonlinear Phenomena, 42(1-3):385–395, 1990.
>
> [2] Liu, J., Lian, J., Sprott, J. C., Liu, Q., and Ma, Y. The butterfly effect in primary visual cortex. IEEE Transactions on Computers, 71(11):2803–2815, 2022.
>
> **A2:** The period $k$ has no physical relationship with the event stream (such as velocity or direction) and cannot be used to improve classification performance. The derived period $k$ is used to demonstrate that the CCNN generates periodic signal outputs when subjected to constant stimuli.
>
> **A3:** The Gaussian wavelet is widely used in time-frequency analysis. Compared to the Morlet wavelet, it offers superior smoothness and better time-frequency localization, making it particularly suitable for analyzing periodic and chaotic signals generated by CCNN neurons.
>
> **A4:** This design does not lead to information loss. Setting negative values to 255 is intended to extract the coordinates of event signal points with constant polarity, thereby detecting moving objects. A more refined threshold design will be presented in future papers.
>
> **A5:** As event density increases, the proposed framework captures finer textures, improving IoU. While denser events enhance performance, the framework is not dependent on them. For sparse events, effective denoising and enhancement techniques may be needed to optimize IoU.
>
> **A6:** Our model uses ResNet-34 with 21.9M parameters, more than EV-VGCNN (0.8M), but parameter count mainly affects memory usage and training time, not inference speed. The faster inference is due to: (1) Efficient 3×3 convolutions and residual connections optimizing computation flow; (2) GPU-optimized deep learning frameworks (cuDNN) accelerating ResNet inference; (3) Convolutional optimizations like Winograd transformations reducing computational complexity. In contrast, EV-VGCNN's fully connected layers may limit parallelism, leading to slower inference.
>
> **A7:**  The LIF method suppresses many event points due to an unreasonable threshold, leading to an incomplete event pattern. The Time Surface captures motion trajectories but introduces redundancy, affecting IoU detection. Event Count maps event accumulation to pixel values, causing inconsistencies that impact pattern integrity. This paper proposes a framework that converts events with constant and periodic polarity changes into periodic and chaotic signals via CCNN, then processes them using CWT and LPF. Event coordinates with constant polarity are set to 255 to extract valid event signals effectively.
>
> **A8:** Dataset size and category count significantly impact classification performance. N-MNIST, N-CARS, and ASL-DVS, with fewer categories and high intra-class similarity, enable effective feature learning and high accuracy. In contrast, N-Caltech101's limited samples, diverse categories, and large intra-class variation increase training difficulty and lower accuracy. This study introduces a chaotic dynamic framework inspired by the dorsal visual pathway, generating robust event representations. Combined with lightweight ResNet34, it achieves high accuracy across all four datasets.
>
> **A9:** The appendix and open-source code provide the full CCNN implementation, including network architecture and hyperparameters. This documentation enables readers to reproduce results with high confidence in their reproducibility.

---

### Decision · Program_Chairs · 2025-05-01

**Decision:**

Accept (poster)

**Comment:**

This paper proposes a chaotic dynamics-based framework inspired by the dorsal visual pathway for processing event camera signals. Using Continuous Coupled Neural Networks (CCNN) and Continuous Wavelet Transform (CWT), the method encodes event sequences as periodic or chaotic signals and achieves strong performance on multiple event classification tasks. According to the reviewers, the paper integrates theoretical modeling, biological inspiration, and experimental validation effectively.
The proposed method demonstrates strong generalization, computational efficiency, and state-of-the-art accuracy across multiple datasets. The framework introduces a novel perspective by leveraging chaos theory and brain-inspired dynamics, which reviewers believe adds both theoretical and practical value. The open-sourced code and comprehensive experiments can also further enhance the paper’s reproducibility and credibility.

Some reviewers also pointed out that the biological connection to the dorsal stream might be weakly supported. According to reviewers, there are also missing analyses on CCNN parameter sensitivity, unclear variable definitions in equations, and limited dynamic system analysis (e.g., no Lyapunov exponent). Authors may want to further enhance the paper by addressing these aspects.